# Career Calling and Job Satisfaction of Ideological and Political Education Teachers in China: The Mediating Role of Occupational Self-Efficacy

**Weiwei Shang [1,*], Guwen Zhang [2,†] and Yanlong Wang [3,*]**

1   School of Marxism, East China Normal University, Shanghai, 200241, China
2   Faculty of Education, East China Normal University, Shanghai 200241, China
3   Marxism Department, Northeast Normal University, Changchun 130024, China
*   Correspondence: wwshang@mks.ecnu.edu.cn (W.S.); wangyl321@nenu.edu.cn (Y.W.)
†   These authors contributed equally to the first author's work.

**Abstract:** Analyzing job satisfaction could help to improve the subjective wellbeing of society. Although the topic of job satisfaction has aroused considerable scholarly interest, research has yet to determine how it relates to career calling and occupational self-efficacy. Drawing on social cognitive career theory, this study aimed to ascertain how career calling influences job satisfaction among ideological and political education teachers, based on the mediating role of occupational self-efficacy. Using the Structural Equation Modeling of survey data from 536 ideological and political education teachers in China, we found that career calling significantly influenced job satisfaction, and occupational self-efficacy was a key mediator between career calling and job satisfaction. These results may inform both improvements to teacher training programs and educational management initiatives to raise the levels of teachers' job satisfaction.

**Keywords:** career calling; occupational self-efficacy; job satisfaction; social cognitive career theory; ideological and political education teachers

## 1. Introduction

A growing body of literature highlights the importance of teachers' job satisfaction. Given the increasing abundance of talent today, job satisfaction is an important indicator of the ability to attract and retain talent to become a teacher, and to stimulate innovation in the teacher community. Job satisfaction plays a key role in human career sustainable development and perceived well-being [1]. It has been defined as "a pleasurable or positive emotional state derived from the appraisal of job experience" [2]. Job satisfaction is a core component of general wellbeing at work. It denotes one's attitude towards the workplace [3], and is fundamental to evaluating team stability [4,5]. Due to the shortage of teachers, their job satisfaction has recently attracted a great deal of attention both nationally and internationally [6,7]. Job satisfaction denotes how teachers relate emotionally to educational work and how they perceive the outcomes of teaching tasks [8]. In this study, we defined job satisfaction as "a state of mind determined by the extent to which the individual perceives her/his job-related needs to be met" [9].

Career calling (CC) is a hot topic in occupational psychology, and it has a positive impact on individual job satisfaction and career development [10,11]. CC, considered to be the highest level of subjective career success, can be what someone views as their purpose in life [12]. People are also likely to feel their involvement in the calling domain is meaningful—that is, that their involvement is "experienced as particularly significant and holding more positive meaning" than other activities [13]. This is significant because meaningfulness itself is a fundamental part of psychological wellbeing, happiness, and

satisfaction [14]. The literature has consistently linked CC to positive work outcomes and experience.

In China, the ideological and political education course is a key course in implementing the fundamental task of educating people with moral values in higher education. Ideological and political education (IPE) is taught to all university students. IPE teachers teach Marxist ideological theory and politics to students, and they are, therefore, central to the developmental aims of higher education in China. To paraphrase Yang and Li [15], the mission of IPE teachers is to nurture talent for the nation. The centrality of IPE teachers to the academy makes their job satisfaction an issue worthy of attention from researchers and other practitioners. Research on the job satisfaction of IPE teachers is a strong basis for examining the stability of IPE teachers and promoting professional development. Previous studies have demonstrated that teachers' job satisfaction is linked to higher retention rates [16]. Furthermore, teachers' dissatisfaction with their job will have a direct impact on their teaching behavior. University administrators must, therefore, strive to promote it wherever possible. In Chinese culture, the teacher is "the child's cow," or "the human ladder"; epithets that point to the value attached to the profession and the likelihood that practitioners will consider it a calling rather than ordinary work. These complimentary sayings devoted to teachers reflect the importance of CC to the culture in China. CC, as an important influencing factor in job satisfaction, has been researched in Western contexts. However, there has been little research into the relationship between career calling and job satisfaction among teachers in the Chinese context. In particular, there are few empirical studies on CC and job satisfaction of IPE teachers in higher education, so it is necessary to expand the research results.

According to social cognitive career theory [17], occupational self-efficacy plays an important role in people's career development, and it is the key mechanism for individuals to exert their initiative. SCCT places special emphasis on the impact of occupational self-efficacy on people's occupational performance and professional experiences [18]. Some studies have found a direct positive effect of CC on students' occupational self-efficacy [19]. In addition, occupational self-efficacy will affect the individual's career experience, which allows the individual to obtain more career satisfaction [20]. Yet, there are few empirical studies on occupational self-efficacy and job satisfaction among IPE teachers.

In summary, understanding more about how CC shapes IPE teachers' job satisfaction requires empirical research to reveal how CC contributes to teachers' job satisfaction and thus to the stable development of the IPE teaching force. To address this, according to Social Cognitive Career Theory (SCCT), this study investigated the relationship between CC and job satisfaction, and the mediating effects of occupational self-efficacy among a sample of IPE teachers in China.

## 2. Theory and Hypothesis

### 2.1. Career Calling and Job Satisfaction

Scholars have defined CC as finding personal fulfillment in an individual's work and perceiving it as meaningful and purposeful [12,21]. The concept of career calling is now a prominent part of management theory and practice, and it may enrich the understanding of many organizational phenomena [22]. Understanding career calling is an important step in addressing the question of how individuals seek and derive meaning from their work [23]. Conceptually, it provides a "more humane and meaningful" way for people to understand their own work lives [24]. As a meaningful passion for one's work, CC better predicts individuals' health and growth [10,25,26] and it is closely associated with numerous positive outcomes [27]. It is also positively associated with increased life satisfaction, and better health [28]. Among adults, CC can enhance employees' work commitment and the extent to which they find work meaningful [29]. People with CC bring their core strengths to the job, which is a major source of job satisfaction [30]. CC motivates people to achieve self-actualization and has a healthy positive impact on individual development while promoting a sense of meaningfulness and satisfaction in one's work [31–33]. In this study,

career calling was defined as a meaningful passion people experience with their work [34]. For example, feeling that they were destined to carry out this type of work and they could not imagine doing anything else [35].

Previous studies have already established that career calling and psychological adjustment—represented by higher levels of job and life satisfaction, self-concept clarity, and lower levels of depression—are linked [10,36]. Similar associations exist between CC and positive work attitudes based on indicators such as enthusiasm and work commitment [37]. Moreover, CC is positively associated with job satisfaction and variables such as the meaningfulness of life and life satisfaction [11,38]. The meaningfulness of work and life was also strongly associated with CC among people expressing highly professional identities [34,39,40]; while positive work attitudes have been shown to increase job satisfaction [41]. Lee et al. reported a significant positive effect of CC on job satisfaction in a sample of 328 Korean teachers [42]. The research above was conducted outside China, but it led us to hypothesize that CC would also exert a positive effect on the job satisfaction of IPE teachers in the country.

**Hypothesis 1 (H1).** *CC is positively associated with job satisfaction among IPE teachers.*

*2.2. Occupational Self-Efficacy as a Mediator*

Occupational self-efficacy refers not only to the self-efficacy in the professional context but also to the self-efficacy in the career behavior process, such as career exploration, career problem solving and career decision-making [43]. Bandura's study noted that people with higher occupational self-efficacy are more confident in their careers, and can engage in more positive occupational behaviors and experiences [44]. Teachers' self-efficacy is a subjective perception and judgment of teachers' ability to: influence the value of education, their own level of education and teaching, and the development of individual learning [45], which includes teacher efficacy in teaching strategies, classroom management, and student engagement [46].

According to Hall and Chandler's model [12], CC is an attitude acquired through a combination of economic, social, cultural, family factors, and other professional attitudes. A stronger CC will be reflected in individuals' understanding of their work as being a purposeful activity and they feel more confident in accomplishing a task. CC can help individuals maintain a positive and optimistic attitude toward professional development and behavior, strengthening their identification with their work [47,48]. It has been found to reduce employee cynicism by making work more meaningful and engaging in times of uncertainty [35]. Teachers who viewed their job as a calling expressed the desire to teach for longer, they were more aware of the positive social components of their work, and they were more willing to make job-related personal sacrifices [37,49]. In the framework of SCCT, CC as a learning experience has a significant impact on an individual's occupational self-efficacy [50]. A series of empirical studies with college students also showed that college students with CC often had higher occupational self-efficacy [51,52]. Therefore, the second hypothesis is based on these findings.

**Hypothesis 2 (H2).** *CC is positively related to occupational self-efficacy among IPE teachers.*

Job satisfaction can reflect not only the choice of career goals for teachers but also their perceived evaluation of their chosen career goals. SCCT maintains that individuals with high occupational self-efficacy tend to be more confident, motivated, and actually more likely to achieve their goals as a result of achieving what they want, so they often exhibit high levels of job satisfaction [53,54]. Related studies also show that occupational self-efficacy has a significant positive effect on job satisfaction [20,55,56]. Based on this, we hypothesized that occupational self-efficacy would influence relationships between CC and job satisfaction among IPE teachers.

**Hypothesis 3 (H3).** *Occupational self-efficacy mediates the relation between CC and job satisfaction among IPE teachers.*

### 3. Materials and Methods

*3.1. Data Collection and Participants*

Data were collected by sending links to an online questionnaire survey platform (Wen Juanxing) via Wechat and email. A random sampling strategy supported the online distribution of the questionnaire, which was voluntarily completed by 580 IPE teachers. After the removal of missing data, 536 valid responses remained, with a recovery rate of 92.4%. The IPE teachers' sample comprised women (69.3%) and men (30.7%), while the mean age of all participants was 32.68 years (SD = 7.56). Regarding the participants' level of education, 29.5% of the IPE teachers had a doctoral degree, 45.2% had a master's degree, and 25.3% had a bachelor's degree. From the perspective of teaching experience, there were 287 teachers with five years or less of teaching experience (53.5%), 172 teachers with 6–10 years of teaching experience (32.1%), and 77 teachers with more than 10 years of teaching experience (14.4%).

*3.2. Measures*

3.2.1. Career Calling

CC was assessed using the 12-item Calling Scale [34], reliability and validity of which was confirmed by Lv et al. [57]. The scale also has good reliability in the Chinese cultural context [58]. A Likert 5-point scale was used (from 1 for "strongly disagree" to 5 for "strongly agree"), and asked respondents to rate their agreement with such statements as "I am passionate about being an IPE teacher", "I enjoy being an IPE teacher more than anything else", and "Being a teacher is a deeply moving and gratifying experience for me", with higher scores representing the higher career calling of IPE teachers. In this study, the internal consistency of our research was found to be strong (=0.91).

3.2.2. Job Satisfaction

We assessed job satisfaction with the three-item Job Satisfaction Subscale from the Michigan Organizational Assessment Questionnaire [59], revised and translated into Chinese by Liu et al. [60]. A Likert 5-point scale was used (from 1 for "very dissatisfied" to 5 for "very satisfied"), and asked respondents to rate their agreement with such statements as "In general, I don't like my job" and "All in all, I am satisfied with my job", with higher scores representing the higher job satisfaction of IPE teachers. In this study, the internal consistency was confirmed by a high alpha value (=0.89).

3.2.3. Occupational Self-Efficacy

Occupational self-efficacy was assessed using the Teachers' Sense of Efficacy Scale Short Form by Tschannen-Moran and Hoy [61]. The scale includes three dimensions with 12 items: efficacy of teaching strategies, classroom management efficacy, and student engagement efficacy. A Likert 5-point scale was used (from 1 for "strongly disagree" to 5 for "strongly agree"), and asked respondents to rate their agreement with such statements as "'I feel prepared for most of the demands of my job", and "I can remain calm when facing difficulties in my job because I can rely on my abilities", with higher scores representing the higher occupational self-efficacy of IPE teachers. In this study, it also had sufficient internal consistency (=0.87).

*3.3. Data Analysis*

The data analysis procedure began by computing the main descriptive statistics and correlations between all variables using SPSS Version 25.0 (IBM, Armonk, NY, USA). The data fit of the model was then evaluated and the path coefficients were estimated using AMOS 25.0 (IBM). Next, we estimated one mediation model with gender as a control variable using confirmatory factor analysis (CFA) to verify the relationship between the

three latent variables of CC, occupational self-efficacy, and job satisfaction. The structural equation model (SEM) was then tested using maximum-likelihood estimation (MLE). The level of statistical significance level was set at $p < 0.05$, and the thresholds for model fit (CFA) were as follows: a root mean square error of approximation (RMSEA) of <0.06, a standardized root means square residual (SRMR) of <0.08, a comparative fit index (CFI) of >0.95, and a Tucker Lewis index (TLI) of >0.95, at a 90% confidence interval [62,63].

## 4. Results

### 4.1. Correlation Analysis of Main Variables

The study investigated all relationships between the variables. The Pearson correlations analysis of the means and SDs of these variables are presented in Table 1. As the Table indicates, CC was positively correlated with occupational self-efficacy (r = 0.565, $p < 0.01$) and job satisfaction (r = 0.319, $p < 0.01$). Moreover, occupational self-efficacy was significantly correlated with job satisfaction (r = 0.463, $p < 0.01$). We therefore inferred that CC and occupational self-efficacy were both positively correlated with job satisfaction.

**Table 1.** Means, standard deviations, correlations, and reliabilities.

| Measures | M $\pm$ SD | 1 | 2 | 3 |
|---|---|---|---|---|
| 1. CC | 3.49 $\pm$ 0.768 | 1.000 | | |
| 2. Occupational self-efficacy | 3.88 $\pm$ 0.641 | 0.565 ** | 1.000 | |
| 3. Job Satisfaction | 3.46 $\pm$ 0.716 | 0.319 ** | 0.463 ** | 1.000 |

** $p < 0.01$; CC, career calling.

### 4.2. The Mediating Effect of Occupational Self-Efficacy

Based on the conceptual model, structural equation modeling (SEM) was carried out using AMOS software to explore the role of CC and occupational self-efficacy on job satisfaction as latent variables. We controlled for gender by connecting it to the endogenous variables and running a series of path analyses. The indices demonstrated that the model provided a satisfactory fit to the data: $X^2/df$ = 2.72, CFI = 0.989, TLI = 0.983, RMSEA = 0.051, and SRMR = 0.014. Figure 1 shows that all proposed paths in the model were significant at the 0.05 level or above. CC had a significant positive effect on occupational self-efficacy (β = 0.546, $p < 0.001$) and job satisfaction (β = 0.479, $p < 0.001$). At the same time, occupational self-efficacy (β = 0.572, $p < 0.001$) exerted a significant positive effect on job satisfaction. Hypotheses 1 and 2 were thus verified.

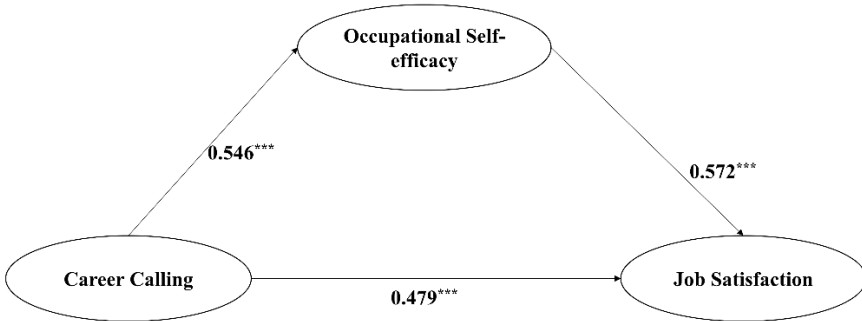

**Figure 1.** The mediation model with occupational self-efficacy as a mediator of the linkage between career and job satisfaction. *** $p < 0.001$.

We then tested whether occupational self-efficacy played a mediating role in CC and job satisfaction using a 5000 bootstrap sample. Table 2 shows that the partial mediating effect of occupational self-efficacy was 0.312, with a 95% CI [0.136, 0.471], excluding 0, thereby supporting hypothesis 3. The mediating effects accounted for 39.3% of the total effect.

**Table 2.** Bootstrap analyses of the significance of mediation (controlling for gender).

| Model Pathways | Effect Size | 95% Confidence Interval | | Percentage |
|---|---|---|---|---|
| | | Boot LLCI | Boot ULCI | |
| CC→JS (direct effect) | 0.479 *** | 0.111 | 0.262 | |
| CC→OSE→JS (indirect effect) | 0.312 *** | 0.136 | 0.471 | 39.4% |

*** $p < 0.01$; CC, career calling; OSE, occupational self-efficacy; JS, job satisfaction.

## 5. Discussion

Drawing on previous research, this study provides a theoretical framework to explain the mechanism by which CC affects job satisfaction among IPE teachers. The study shows that the job satisfaction of these IPE teachers benefited from their sense of CC.

### 5.1. Theoretical Implications

We found that the relationship between CC and job satisfaction was partially explained by the teachers' level of occupational self-efficacy, after controlling for gender. Our results also indicated that IPE teachers with a high CC were more likely to feel satisfied with their job through occupational self-efficacy. Consistent with earlier CC research [64], employees who saw their work as a calling put more effort into their careers and it made them feel more confident when facing problems at work, thereby increasing their satisfaction. However, CC does not necessarily lead to job satisfaction unless individuals believe in and are committed to their own professional development [65]; moreover, occupational self-efficacy further provides the conditions for individuals' well-being and satisfaction in work. Our study showed IPE teachers who were likely to see teaching as a calling were also most satisfied with their work. While these findings apply to Western settings, the study makes a valuable contribution to literature based on the Chinese context. In China, IPE teachers have the mission to nurture people for the Communist Party, promote socialist core values and to nurture talents for the country [66]. The work of IPE teachers is the work of enriching life and creating life; teachers are the engineers of human souls in China. Additionally, IPE teachers must be loyal, love their profession be selfless and dedicated.

Occupational self-efficacy is an important mechanism, which is consistent with SSCT's view [17]. The final result demonstrated that occupational self-efficacy was also a critical factor in the relationship between CC and job satisfaction, which enriched the understanding of the predictive effect variables of job satisfaction. This study provides an in-depth theoretical explanation of the intrinsic relationship between CC and job satisfaction in the framework of SSCT. In addition to guidance on ideals and values in IPE teachers' training [67], career development counseling should be enhanced to improve IPE teachers' occupational self-efficacy. This is in line with other research showing that career calling as an occupational attitude can have significant effects on job outcomes [19,38,65].

### 5.2. Practical Implications

The above findings provide educational administrators with a wealth of practical insights. Teacher training needs to shift from being dedicated to cultivating IPE teachers' teaching skills to cultivating educational calling, guiding them to establish noble moral pursuits and professional sentiments, and leading the cultivation work with a strong internal drive. Maintaining a strong sense of CC and occupational self-efficacy can help to sustain satisfaction and well-being throughout the whole career. It is important to focus on stimulating or cultivating CC of IPE teachers, helping them to clarify their understanding of the teaching profession, and guiding them to associate the profession with their personal sense of meaning or value in life, making their lives more meaningful, so that they can truly believe and actively engage in the teaching profession. At the same time, education management can cultivate IPE teachers' occupational self-efficacy, such as the success of the experience. We can help teachers integrate their past life experiences with their

current feelings and future expectations and strengthen their professional beginnings and occupational self-efficacy through the distillation of life themes, thus promoting job satisfaction and sustainable development.

### 5.3. Limitations and Future Study

Several limitations of this study should be mentioned. First, its focus on the mediating mechanisms of CC and job satisfaction excluded other factors, such as job performance [38]. Further research should consider such factors as additional mediating or moderating variables to better understand the relationship between CC and job satisfaction, and it should consider more control variables. Second, because the sample in the present study was collected from mainland China, further research should examine other contexts. Furthermore, in this paper the absence of differential analysis, such as country, gender, age categories, and years of experience, is a drawback. A comprehensive comparative analysis using the analysis of variance and other methods will be dedicated in future studies. Third, the cross-sectional study design prevented us from understanding the temporal precedence among CC, occupational self-efficacy and job satisfaction, a problem that could be solved via a longitudinal study.

### 6. Conclusions

This study aimed to identify the direct and indirect predictive factors influencing the job satisfaction of IPE teachers in mainland China. It provides an important preliminary understanding of how the career calling of IPE teachers influences their job satisfaction through occupational self-efficacy. The findings indicate that teacher trainers should cultivate career calling, and teachers should aim to improve occupational self-efficacy to increase their levels of job satisfaction.

**Author Contributions:** All authors listed have made a substantial, direct, and intellectual contribution to the work, and approved it for publication. All authors have read and agreed to the published version of the manuscript.

**Funding:** This study was supported by The National Social Science Fund of the Chinese government [No.21CKS007].

**Institutional Review Board Statement:** The study was conducted in accordance with the Declaration of Helsinki, and approved by the Ethics Committee of East China Normal University (HR 373-2022).

**Informed Consent Statement:** Informed consent was obtained from all subjects involved in the study.

**Data Availability Statement:** The raw data supporting the conclusions of this article will be made available by the authors, without undue reservation.

**Conflicts of Interest:** The authors declare no conflict of interest.

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
