# Peer review of "Career Calling and Job Satisfaction of Ideological and Political Education Teachers in China: The Mediating Role of Occupational Self-Efficacy"

_sustainability, doi:10.3390/su142013066_

Round 1
Reviewer 1 Report
This is a good mainstream article presenting evidence from Chinese teachers, an understudied population. It is reasonably well-written, and the methods are clear.
I would ask the authors to develop more clearly the methods for data sampling, collection, and processing. They state that the sample is random, could you please explain what was the population you sampled from? Is there a public list of teachers you used as a sampling framework? Please provide more information about the model of exclusion; how many missing items the removed cases had (considering there were 44 excluded cases)? was there a protocol for that? was any imputation procedure used?
Finally, regarding the analysis, it would be useful to test for invariance between males and females, particularly because work usually involves an extra effort by women, and therefore, the role of self-efficacy could be even higher for them.
Author Response
Thank you very much for your suggestions.
The data sample was randomly sampled in different cities, involving central, eastern, and western cities in China, such as Shanghai, Shandong, and Shaanxi, etc. The sample subjects were IPE teachers at universities.
Based on research ethics, the sample subjects in this study are anonymous, and the teachers' names are not disclosed.
The number of missing items in each case of deletion is inconsistent, but all have the problem of some unanswered items. There is no specific agreement, deletion is a way to conduct data analysis, form now on other interpolation procedures are not used in this study, in future studies are prepared to consider the use of interpolation methods such as mean fill method.
It is a good suggestion that gender differences will be studied specifically in future studies and this will be placed as a limitation part of this thesis. In the future, the study will provide a more comprehensive analysis of gender differences.
Reviewer 2 Report
1. More details of participants should be provided. In the (Data collection and Participants). For example male vs. female, age categories, years of experience, etc.
2. Some analysis of variance might better shed light on the quality of the paper. Such analysis should focus on the different categories of the participants. The absence of such analysis is a major drawback in this paper.
3. Much more elaborations are needed on the (items) in each of the dimensions in the structural equation model. The literature review should be expanded to elaborate more on the (items) in each dimension.
4. The discussion part should be expanded more to better connect results with the (Chinese) culture and environment,
5. Reference list should be updated. It should also focus more on more recent literature on similar Chinese cultures.
6. Little discussions on the (direct effects), (indirect effects), and (total effects) of the final model might be recommended.
Author Response
Thank you very much for your suggestions.
1.The article refines more details of participants, including male vs. female, age categories, education degree, years of experience,etc.
2.It is a good suggestion that future studies will specifically analyze the difference of gender, years of experience, age categories, education degree, etc, using ANOVA and other analysis methods.
3.The article is supplemented with related literature.
4.The discussion section adds how to promote the sustainability and well-being of IPE teachers in the professional field in the context of Chinese culture and environment.
5.The article is supplemented with relevant references and discussion.
6.In part of Theoretical Implications, the article discuss direct effects, indirect effects and total effects.
Reviewer 3 Report
The paper addresses the relationship between career calling and job satisfaction mediated by occupational self-efficacy in a sample of teachers from China. Understanding job satisfaction in teaching and its antecedents is a theoretically interesting and practically important topic.
Below, please find my concerns regarding the manuscript and some suggestions on how this manuscript could be improved. These regard mostly the introduction and discussion parts:
1. I see the introduction as less convincing because it is actually too descriptive. The main message is somehow muddled with definitions. I would advise the authors to reorganize it and put a great emphasis on the gap the paper addresses.
2. The specific characteristic of the sample is tackled in the introduction, but with no other mention throughout the paper. It would be important to follow such a thing given that it is primed by the title.
3. The discussion section is framed following the well-known guidelines:
i. Main findings of the present study;
ii. Comparison with other studies;
iii. Implication and explanation of findings;
iv. Strengths and limitations.
However, I see it necessary to enrich the mirroring of the results to those from previous studies taking into account the context of China. Apart from that, the results are crystal clear with little contribution to the literature.
4. Sample items for the instruments of data collection should be offered.
5. Why Spearman correlation?
6. How gender was tested as a control variable? Please mention other control variables that could be relevant.
4. The quality of the writing is a bit variable throughout the paper. Some parts are less clear than others and the paper may benefit from general improvements in terms of clarity and a more consistent writing style.
- lines 24-27 - the sentence is too long. It should be split to make easier this understanding.
- line 117 - teaching as a career would be deleted
- line 163 but not only: it is Likert scale
I wish the authors good luck with further developing the paper.
Author Response
Thank you very much for your suggestions.
1.The article improved the introduction section.
2.The article adds refinements to the specific characteristics of the sample within the context of the full article.
3.The discussion section was supplemented with how the Chinese cultural and environmental context promotes sustainable development and well-being of IPE teachers in the professional field. in lines 269-272, 305-308.
3.In the part of Measures, sample items were added.
4.Pearson correlation analysis was used as an assessment of correlations between variables and as a prerequisite for structural equation modeling.
5.In this thesis, gender which is important as interfering factor was used as a control variable and brought into the structural equation model in the form of dummy variables for testing, and variables such as education and age will be included in the control variables in future studies.
6.The writing language of the paper was improved,in lines 23-27, 183,192,201.
Round 2
Reviewer 3 Report
I have read the revised version of the manuscript and I see that the authors have addressed the comments and suggestions so that the text has been improved.
However, I still noticed a discrepancy between the authors' response and the manuscript regarding the correlation test. I would ask for correction.
Author Response
Thank you very much!It does need to be modified.
The article was revised. It should be a Pearson correlation analysis.